# Effect of Process Parameters on Interfacial Bonding Properties of Aluminium–Copper Clad Sheet Processed by Multi-Pass Friction Stir-Welding Technique

**Nora Osman, Zainuddin Sajuri \*, Amir Hossein Baghdadi and Mohd Zaidi Omar**

Centre for Materials Engineering and Smart Manufacturing (MERCU), Faculty of Engineering and Built Environment, Universiti Kebangsaan Malaysia, Bangi 43600 UKM, Selangor, Malaysia; nora@siswa.ukm.edu.my (N.O.); baghdadi.amirhossein@gmail.com (A.H.B.); zaidiomar@ukm.edu.my (M.Z.O.)
\* Correspondence: zsajuri@ukm.edu.my; Tel.: +60-3-8911-8017

**Abstract:** In this study, continuous multi-pass friction stir welding was used to clad dissimilar AA6061 aluminium (Al) and C2801P copper (Cu) alloy materials. The empirical relationships between three process parameters and two-factor responses of Al–Cu clad joints were evaluated. Mathematical models were generated using regression analysis to predict the variation in tensile shear and peel load of the cladded joints. The sufficiency of the developed model was validated by analysis of variance (ANOVA), and the multi-criterion optimisation of factor responses was carried out via the response surface method. Results showed the formation of mechanical interlocking at the cladded interface and the development of a thin metallurgical bonding layer consisting of Al alloy content (8–21%), which greatly affected the quality of the Al–Cu joint interface. Moreover, the increase in shoulder overlap ratio, welding speed and tool rotational speed improved the shear and peel strength up to a certain range before gradually declining. The optimised process parameters for the cladded Al–Cu were obtained at a rotational speed of 986 r/min, welding speed of 8.6 mm/min and shoulder overlap ratio of 35%. The cladded Al–Cu generated a shear strength of 5850 kPa and peel strength of 750 kPa with an overall desirability function of 0.94.

**Keywords:** multi-pass friction stir welding; cladding; dissimilar material; aluminium; copper; response surface method

## 1. Introduction

Copper (Cu) and aluminium (Al) metallic alloys have broad applications in heating, ventilation and air conditioning, refrigeration and electrical industries due to their properties, such as high electrical and thermal conductivity and corrosion resistance [1–4]. Cu alloy is a relatively expensive material with a higher density compared with Al alloy [5]. Therefore, industries often substitute Cu-based components with Al to reduce overhead costs [6,7]. The full substitution of Cu with Al is not feasible for most industrial applications because it affects the overall manufacturing; thus, the fabrication of Cu–Al joints in related industries is in high demand [8,9]. The joining technique of Cu and Al metals is challenging due to differences in their metallurgical and mechanical properties, such as intermetallic bonding, cracks, and cavities [10,11].

Solid-state welding methods such as ultrasonic welding [12,13], accumulative roll bonding [14,15], and explosive cladding [1,16,17], are feasible methods for joining Al to Cu because of the use of low heat input required for joining. However, these methods have several disadvantages. For example,

ultrasonic welding and accumulative roll bonding methods lack versatility, and explosive cladding has a safety issue problem during application [18].

Recently, much attention has been drawn to the friction stir welding (FSW) of dissimilar materials due to its various advantages, such as low temperature, low residual stress, versatility, environmental friendliness and no need for filler materials [19–26]. In FSW, a non-consumable rotating tool with a specific designed shoulder and pin is plunged into the plates to be joined and traversed along the line of joint. The tool heats the workpiece and moves the material to produce the joint [27]. Heating is accomplished by friction between the tool and the workpiece and plastic deformation of the workpiece. Joining is achieved by the simultaneous action of the rotating tool and transverse movements that cause heating and soften the material around the pin and shoulder [28].

A number of studies successfully cladded dissimilar materials by using FSW methods [29–31]. Remarkable mechanical properties were obtained in the Al–Cu joints at suitable FSW parameters [32–35]. Although FSW has gained importance, the current literature is still mostly concerned about single-pass laps and butt joints [9,36]. Therefore, the use of the multi-pass strategy by FSW to increase the bonding area in joining dissimilar metal plates is currently gaining interest among researchers. Leitao et al. [37] performed multi-pass friction stir lap welding of Al–Fe and found that no intermetallic compound (IMC) forms in the Al–Fe interface. They proposed that the bonding mechanisms are diffusion bonding and mechanical bonding via a wavy feature. Meanwhile, a multi-pass friction stir cladding study performed by Sorger et al. [38] demonstrated that the overlapped welding of dissimilar Al–Fe joints produces a wave-like shape interface that can improve the mechanical properties of the cladded Al–Fe. Zhao et al. [39] performed multi-pass FSW to clad AZ80 and Al sheets. A composite joint with uniform grains was obtained at the Mg–Al interface with the existence of Mg/Al IMCs in the weld interface. However, a study on multi-pass friction stir clad welding of Al to Cu has not been conducted yet.

In this study, continuous multi-pass FSW was utilised to clad dissimilar AA6061 aluminium (Al) and C2801P copper (Cu) alloys. Three FSW process parameters were considered in this study, namely, shoulder overlap ratio, rotational speed and welding speed. The empirical relationships between the three process parameters and two-factor responses of Al–Cu clad joints were evaluated. The design of experiment (DOE) was employed to systematically analyse the effect of factors on responses [40] that determine the quality of the joined interface. This method uses a mathematical and statistical approach to optimise numerical data analysis. Several authors have employed the Taguchi method [41] and response surface methodology (RSM) [42] in designing their experiment and optimising process parameters in FSW. Although these studies applied mathematical models in FSW, research on the development of a mathematical model for FSW Al–Cu joint parameters and responses is currently lacking. Therefore, in this study, statistical analyses, such as analysis of variance (ANOVA) and RSM, were utilised to predict process factors including the shear and peel strength of the FSW Al–Cu clad. Furthermore, a multiresponse optimisation procedure was used as a tool to identify the most optimal parameter combination that can maximise the shear and peel strength factors for dissimilar Al–Cu joints through the multi-pass overlap strategy by shifting the tool during welding.

## 2. Materials and Methods

### 2.1. Materials Used and Identification of Process Parameters

Aluminium AA6061 and copper C2801P alloys with the dimensions of 150 mm × 150 mm × 1.5 mm were used as the base metals in this study. The chemical compositions of the alloys are listed in Table 1. FSW was carried out using a conventional milling machine, in which the Al alloy sheet was overlaid on a Cu alloy sheet and firmly clamped onto fixtures. Here, the Al alloy sheet is the cladding material and the Cu alloy sheet is the substrate. The welding tool used in this process was a D2 tool steel alloy with a concave shoulder diameter size of 14 mm and a threaded tapered pin with diameter and length of 4 and 2 mm, respectively (Figure 1a). The tilt angle of the rotating tool was fixed at 3° along the z-axis of the milling machine, and the penetration depth was constant at 2.2 mm for all samples, as

illustrated in Figure 1b. For all welding conditions, the Al sheet was always positioned at the top of the Cu sheet. Each weld pass was fabricated using the same tools and welding parameters, as shown in Figure 2, with the tool rotation, welding direction and top sheet kept constant throughout the process. The shoulder overlap ratio was set at 0%, 50% and 100% with the width of the overlap dimension kept constant at 28 mm for all overlap conditions.

**Table 1.** Chemical composition of the welded materials.

| | Chemical Composition [wt%] | | | | | | | | | | |
|---|---|---|---|---|---|---|---|---|---|---|---|
| Alloys | Mg | Si | Cu | Zn | Fe | S | Cl | Cr | Na | Ni | Al |
| AA6061 | 1.00 | 0.91 | 0.30 | 0.02 | 0.21 | 0.11 | 0.09 | 0.08 | 0.07 | 0.01 | Balance |
| C2801P | - | - | 60 | 40 | - | - | - | - | - | - | - |

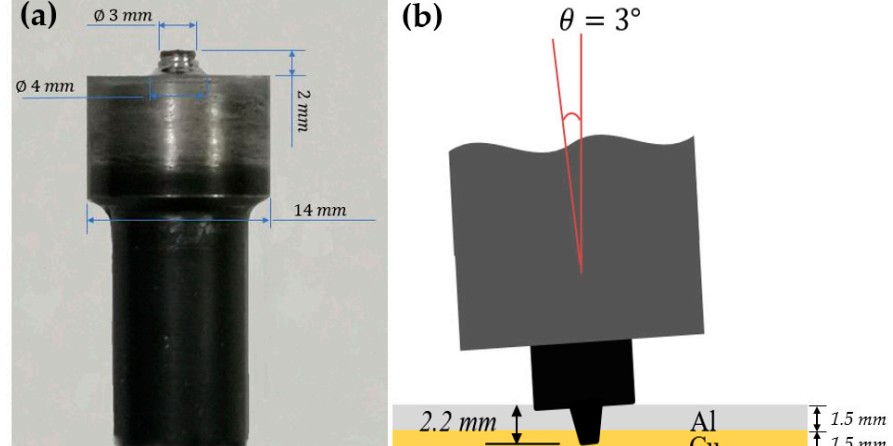

**Figure 1.** Schematic diagram of (**a**) tilt angle and plunge depth during the welding process and (**b**) the friction stir-welding (FSW) tool.

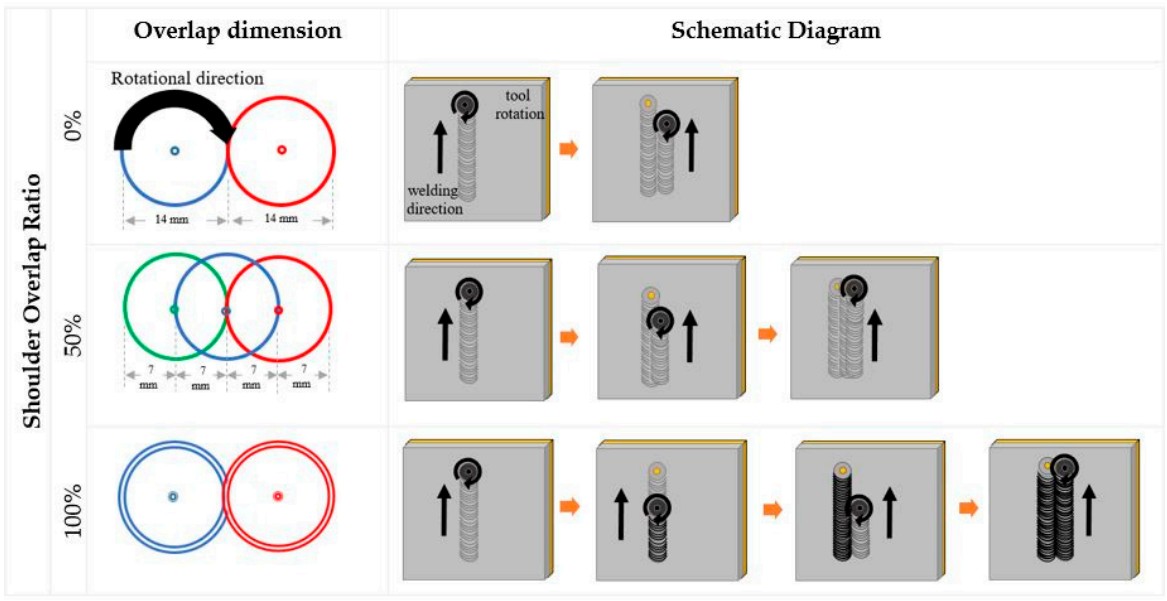

**Figure 2.** Schematic diagram of the shoulder overlap ratio for FSW Al–Cu.

Figure 3 shows the location of the samples for metallographic analyses, shear and peel tests which were machined from the Al–Cu cladded sheet. For metallographic analyses, the samples were prepared according to the standard procedures outlined in ASTM E3-11 (2017). The Cu alloy

microstructure was revealed by a chemical solution containing 10 g of $FeCl_3$ (Merck KGaA, Darmstadt, Germany), 30 mL of HCl (Merck KGaA, Darmstadt, Germany) and 85 mL of $H_2O$, whereas the Al alloy microstructure was revealed using Keller's reagent. Both metal alloys were imaged using an Olympus optical microscope (Olympus Optical Co., Ltd., Shinjuku-ku, Japan). For metallurgical analysis, both field-emission scanning electron microscopy (FESEM) and energy-dispersive X-ray spectroscopy (EDS) were analyzed using ZEISS-Merlin FESEM (Carl Zeiss Microscopy GmbH, Jena, Germany) to perform detail observation on the microstructure, fracture surface and elemental analysis of the joint samples. Furthermore, samples were analyzed by X-ray diffraction (XRD) to identify the presence of intermetallic compounds. XRD analysis was performed using a Bruker AXS, Karlsruhe, Germany, diffractometer with monochromatic Cu-Kα radiation (λ = 0.1541 nm) at 40 kV and 4 mA. The peel and shear tests were conducted to determine the shear stress and peel failure strength of the cladded joint samples.

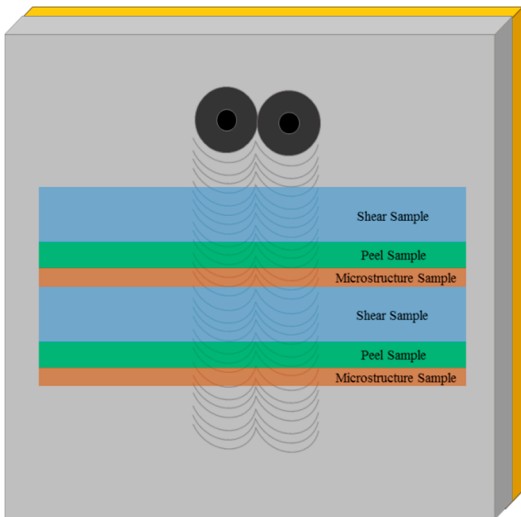

**Figure 3.** Schematic diagram of sample cutting for shear, peel and microstructure analysis.

The dimensions of the sample for the shear and peel test are illustrated in Figure 4. The shear test was performed according to ASTM D1002 (2010) by using the Zwick 100 kN load frame universal testing machine (ZwickRoell, Ulm, Germany) at a constant cross-head displacement rate of 1 mm/min. Peel tests were also carried out as per ASTM D1876-08(2015) at a rate of 1 mm/min. An Accuracy Class 1 load cell of 20 kN (measurement range from 40 N to 20,000 N) was used for load measurement. Three samples were tested for each parameter combination to evaluate the shear and peel strengths of the joints.

## 2.2. Evaluation of the Process Parameters

A feasible study on the operating process parameters was performed to fulfil the welding criteria, in which only defect-free Al–Cu cladded weld joint samples were selected. To determine the optimal working range of the rotational and welding speeds, a preliminary test was conducted by performing a welding pass (no overlap) via a combination of both the rotational and welding speeds. The speeds were set based on the machine specification and capacity, ranging from the minimum to maximum speed level. The feasible speed range limits were identified based on factors, such as visual inspection, plate separation, surface defects, the formation of welding flash, cross-sectional flaws and appearances of the weld bead. The results of the analyses are shown in Table 2.

## 2.3. Design of Experiments

Throughout this study, welding speed, rotational speed and shoulder overlap ratio were denoted as parameters or factors for cladding dissimilar Al–Cu joints using friction stir welding that produces two responses, namely shear strength $\sigma_s$ and dan peel strength $\sigma_p$. The experiments were designed

based on historical data design (as shown in Table 3) in Design Expert Version 6.0 software (Stat Ease, Inc., Minnesota, MN, USA), and the value for the factors was obtained from a feasible study (for rotational and welding speed ranges). The shoulder overlap ratio strategy was carried out to identify the effect of overlap strategy on the strength of the cladded joints. The lower limit of the factor was coded −1, and the upper limit was coded +1. This historical data design matrix involved 27 experimental processes at three independent input variables. Furthermore, the RSM was used to establish an empirical relationship between the FSW input factors and output responses.

The adequacy of the established relationships was verified using ANOVA. ANOVA determines the models' significant value of each process factor in *P*-value (Probability of significance); if *P*-value for a term is smaller than 0.05 (at a 95% confidence level), then it can be summarised that the model is significant on the designated response. If there were more than 2 factors that have *P*-value less than 0.0001, then the *F*-value plays a role to decide the sequence of the most significant factor. The *F*-value or *F* statistic is a value to find out if the means between two or more populations that are significantly different. The *P*-value is decreasing when the *F*-value is increasing. The most suitable model is obtained when the determination coefficient, $R^2$ is approaching 1; justifying that 99% of the variability of the factor that shows the significance of the model and the goodness of fit for the model.

## 3. Results

### 3.1. Analysis of Variance (ANOVA) Analysis and Regression Model

ANOVA analysis generated a quadratic regression model for the shear strength $\sigma_s$ and peel strength $\sigma_p$ response factors. The predicted values and process errors for both responses were calculated, and results are shown in Table 4. The experimental results of the shear and the peel strengths were the average values of three tested samples, respectively. The peel strength test showed a higher error value due to the high variance in the data obtained as compared with the shear test.

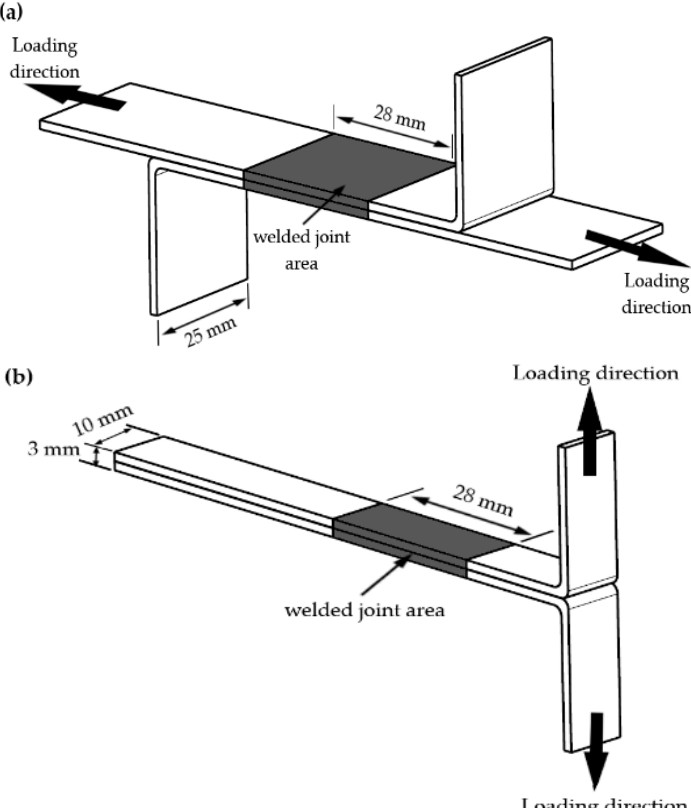

**Figure 4.** The dimension of the samples; (**a**) shear test and (**b**) peel test.

**Table 2.** Inferences from feasible operating limits of welding parameters.

| Parameter | Range | Visual Inspection | Inference |
|---|---|---|---|
| Rotational speed | <825 r/min |  | • Sheet separation due to insufficient heat input<br>• Rough weld beads |
| | >1320 r/min |  | • Extreme weld flash formed, and Cu is extruded to top sheet surface due to excessive heat input<br>• Produce surface groove defect |
| Welding speed | <5.5 mm/min |  | • This is the minimum welding speed<br>• Produce sound weld surface appearance with no defect |
| | >17 mm/min |  | • Formation of surface groove defect (insufficient material filled) and poor Al–Cu bond due to insufficient heat generation because of higher welding speed. |

**Table 3.** Feasible operating limit of FSW Al–Cu.

| Factor | Notation | Level | | |
|---|---|---|---|---|
| | | **−1** | **0** | **1** |
| Rotational speed, (r/min) | $\omega$ | 825 | 1055 | 1320 |
| Welding speed, (mm/min) | $v$ | 5.5 | 9 | 17 |
| Shoulder overlap ratio, (%) | $S_r$ | 0 | 50 | 100 |

**Table 4.** Experimental design matrix with evaluated data.

| Test No. | Rotational Speed, $\omega$ (r/min) | Welding Speed, $v$ (mm/min) | Shoulder Overlap Ratio, $S_r$ (%) | Shear Strength, $\sigma_s$ (kPa) | | | Peel Strength, $\sigma_p$ (kPa) | | |
|---|---|---|---|---|---|---|---|---|---|
| | | | | Actual | Predicted | Error (%) | Actual | Predicted | Error (%) |
| 1 | 825.0 | 5.5 | 0.0 | 4723.9 | 4604.1 | 2.5 | 725.0 | 680.0 | 6.2 |
| 2 | 825.0 | 9.0 | 0.0 | 4342.9 | 4457.7 | 2.6 | 622.1 | 604.3 | 2.9 |
| 3 | 825.0 | 17.0 | 0.0 | 3092.4 | 3048.9 | 1.4 | 275.4 | 296.4 | 7.7 |
| 4 | 1055.0 | 5.5 | 0.0 | 4830.0 | 4803.6 | 0.5 | 500.0 | 536.1 | 7.2 |
| 5 | 1055.0 | 9.0 | 0.0 | 5339.0 | 5328.6 | 0.2 | 751.1 | 710.4 | 5.4 |
| 6 | 1055.0 | 17.0 | 0.0 | 3523.3 | 3534.0 | 0.3 | 287.9 | 295.7 | 2.7 |
| 7 | 1320.0 | 5.5 | 0.0 | 3791.4 | 3840.6 | 1.3 | 428.6 | 414.3 | 3.4 |
| 8 | 1320.0 | 9.0 | 0.0 | 4682.3 | 4365.6 | 6.8 | 352.9 | 374.3 | 6.1 |
| 9 | 1320.0 | 17.0 | 0.0 | 2329.0 | 2413.9 | 3.6 | 106.4 | 109.3 | 2.7 |
| 10 | 825.0 | 5.5 | 50.0 | 5248.6 | 5081.4 | 3.2 | 832.1 | 801.4 | 3.7 |
| 11 | 825.0 | 9.0 | 50.0 | 4701.4 | 4892.1 | 4.1 | 745.4 | 725.7 | 2.6 |
| 12 | 825.0 | 17.0 | 50.0 | 3338.1 | 3383.3 | 1.4 | 350.0 | 360.7 | 3.1 |
| 13 | 1055.0 | 5.5 | 50.0 | 5051.9 | 5138.0 | 1.7 | 657.1 | 714.6 | 8.8 |
| 14 | 1055.0 | 9.0 | 50.0 | 6024.3 | 5843.0 | 3.0 | 785.7 | 725.0 | 7.8 |
| 15 | 1055.0 | 17.0 | 50.0 | 3868.6 | 3868.4 | 0.0 | 402.5 | 417.1 | 3.6 |
| 16 | 1320.0 | 5.5 | 50.0 | 4031.4 | 4175.0 | 3.6 | 583.6 | 535.7 | 8.2 |
| 17 | 1320.0 | 9.0 | 50.0 | 4756.9 | 4700.0 | 1.2 | 428.6 | 460.0 | 7.3 |
| 18 | 1320.0 | 17.0 | 50.0 | 2642.9 | 2701.1 | 2.2 | 357.1 | 330.7 | 7.4 |
| 19 | 825.0 | 5.5 | 100.0 | 1605.7 | 1657.4 | 3.2 | 350.4 | 361.1 | 3 |
| 20 | 825.0 | 9.0 | 100.0 | 2466.1 | 2468.1 | 0.1 | 353.2 | 363.6 | 2.9 |
| 21 | 825.0 | 17.0 | 100.0 | 990.7 | 959.3 | 3.2 | 184.3 | 195.0 | 5.8 |
| 22 | 1055.0 | 5.5 | 100.0 | 2111.4 | 2142.6 | 1.5 | 394.6 | 420.4 | 6.5 |
| 23 | 1055.0 | 9.0 | 100.0 | 2990.0 | 2953.3 | 1.2 | 418.9 | 416.1 | 0.6 |
| 24 | 1055.0 | 17.0 | 100.0 | 1137.1 | 1158.7 | 1.9 | 231.4 | 213.6 | 7.7 |
| 25 | 1320.0 | 5.5 | 100.0 | 1370.9 | 1411.0 | 2.9 | 222.1 | 226.8 | 2.2 |
| 26 | 1320.0 | 9.0 | 100.0 | 2187.1 | 2133.1 | 2.5 | 197.9 | 215.7 | 9 |
| 27 | 1320.0 | 17.0 | 100.0 | 417.1 | 410.0 | 1.7 | 11.8 | 10.4 | 11.9 |

ANOVA statistical analysis also revealed the significance values for both regression models, in which the significance of the generated model and its coefficients were denoted by a large *F*-value and a small *P*-value. As observed in Table 5, the *P* values were lower than 0.0001 for both responses, and the *F*-values of the model were 236.1 and 27.32 for $\sigma_s$ and $\sigma_p$, respectively, thereby confirming the statistical significance of these models. Specifically, the main statistical effects were due to factors such as the rotational speed ($\omega$), welding speed ($v$), shoulder overlap ratio ($S_r$), quadratic effect of rotational speed ($\omega^2$), welding speed ($v^2$) and shoulder overlap ratio ($S_r^2$). On the basis of the *F*-values, the order of significance for $\sigma_s$ and $\sigma_p$ was $S_r > S_r^2 > v^2 > v > \omega^2 > \omega$ and $v > S_r^2 > \omega > S_r > \omega^2 > v^2$, respectively.

The $R^2$ and adjusted $R^2$ values, which specify the adequacy of the developed models, were calculated as 0.9702 and 0.9612 for $\sigma_s$ and 0.8913 and 0.8586 for $\sigma_p$, respectively. A multiple regression analysis-derived mathematical model was generated for both factors, as denoted by the coded factors and represented as follows:

$$\sigma_s = +3960.21 - 165.01\omega - 401.70v - 824.23S_r - 523.76\omega^2 - 862.73v^2 - 1034.87S_r^2 \qquad (1)$$

$$\sigma_p = +190.06 - 27.21\omega - 38.68v - 26.21S_r - 25.15\omega^2 - 26.41v^2 - 60.22S_r^2 \qquad (2)$$

Both Equations (1) and (2) were used to predict if the $\sigma_s$ and $\sigma_p$ of FSW Al–Cu joints were within the optimal welding parameter range. The validation of the models' accuracy was accomplished by plotting the normal probability plot and predicted value versus the actual value plot for $\sigma_s$ and $\sigma_p$ of the FSW Al–Cu joints, as illustrated in Figures 5 and 6, respectively. Figure 5a,b show the errors normally spread for both models as the residual values fell on a straight line. Therefore, Figure 6a,b indicate that the mathematical models for $\sigma_s$ and $\sigma_p$ developed in this study could predict the response values accurately followed the actual experimental response values within the welding parameter range.

**Table 5.** Analysis of variance for two responses.

| Source | Shear Strength, $\sigma_s$ (kPa) | | Peel Strength, $\sigma_p$ (kPa) | |
|---|---|---|---|---|
| | *F*-value | *P*-value, Prob > *F* | *F*-value | *P*-value, Prob > *F* |
| Model | 108.44 | <0.0001 | 27.32 | <0.0001 |
| $\omega$ | 11.37 | 0.003 | 25.07 | <0.0001 |
| $v$ | 80.28 | <0.0001 | 50.64 | <0.0001 |
| $S_r$ | 281.02 | <0.0001 | 23.26 | 0.0001 |
| $\omega^2$ | 33.15 | <0.0001 | 7.06 | 0.0152 |
| $v^2$ | 52.48 | <0.0001 | 5.37 | 0.0312 |
| $Sr^2$ | 153.86 | <0.0001 | 40.91 | <0.0001 |
| $R^2$ | 0.9702 | | 0.8913 | |
| Adjusted $R^2$ | 0.9612 | | 0.8586 | |
| Adequate Precision | 35.72 | | 18.58 | |
| Significant | Yes | | Yes | |
| Lack of fit | Insignificant | | Insignificant | |

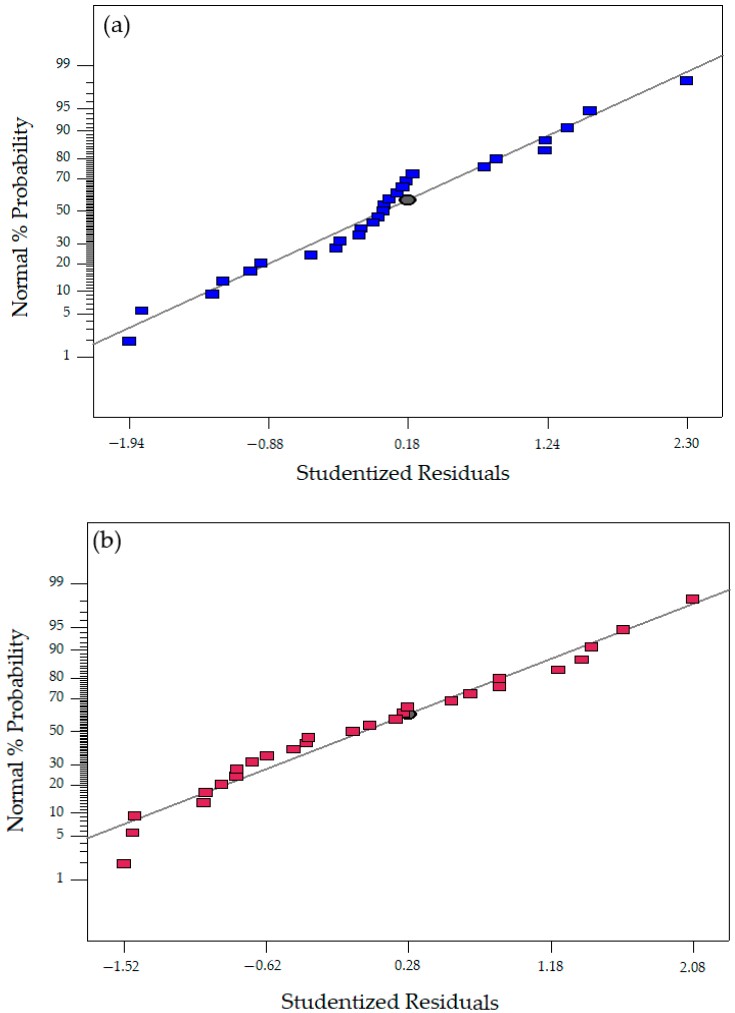

**Figure 5.** The plot of normal probability of residuals: (**a**) shear strength and (**b**) peel strength.

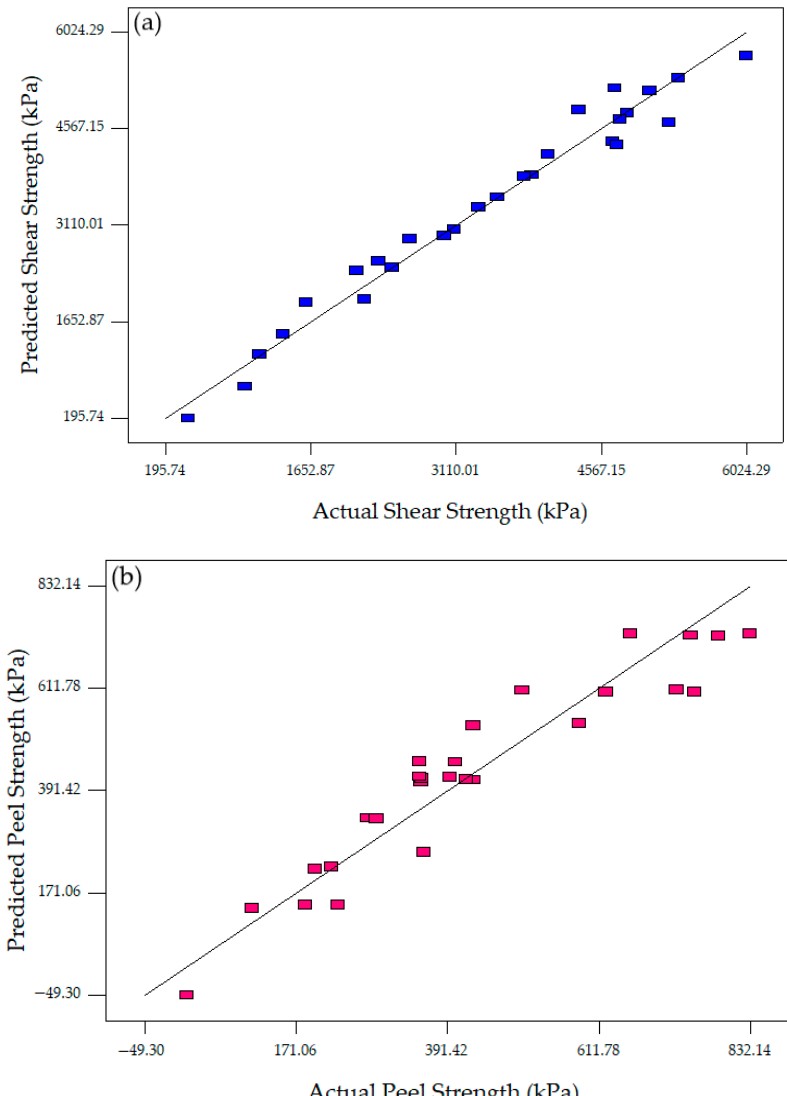

**Figure 6.** The plot of actual vs. predicted response: (**a**) shear strength and (**b**) peel strength.

## 4. Discussions

### 4.1. Effect of Friction Stir-Welding (FSW) Process Parameters on $\sigma_s$ and $\sigma_p$

The fracture load on FSW Al–Cu joints was determined using the tensile shear and peel tests. The effects of each process factor (parameter) were represented in perturbation plots, as shown in Figure 7a,b. These plots display a profile view of the response surface design that indicates how the response factor changes as each FSW parameter shifts from the reference point, with every other factor fixed at a constant reference value. Each parameter was set at a default reference point at the coded zero levels of each factor by the software. When the rotational speed, welding speed and shoulder overlap ratio increased, the tensile shear and peel load factor values also increased. Thus, after reaching the maximum level, the $\sigma_s$ and $\sigma_p$ values decreased with the increase in each factor. These parametric changes caused the development of various frictional heat input that produced different levels of plastic deformation, thereby affecting the strength of the Al–Cu interface [27]. These changes could be visualised using the perturbation plot to determine the effective factors on the response surfaces of the design process. The perturbation plots illustrated in Figure 7a,b show the hierarchy of effective factors for $\sigma_s$ and $\sigma_p$, which are A($\omega$) > B ($v$) > C ($S_r$) and C ($S_r$) > A($\omega$) > B ($v$), respectively.

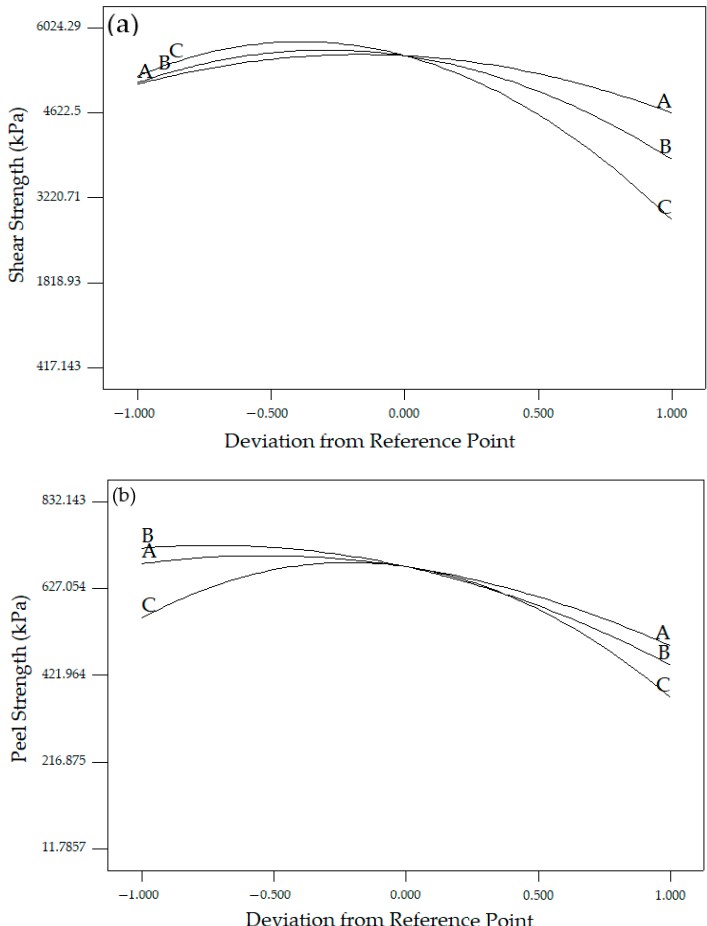

**Figure 7.** Perturbation plots: (**a**) shear strength and (**b**) peel strength.

To achieve optimisation for the two response factors, namely, $\sigma_s$ and $\sigma_p$, Design Expert Version 6.0 software was utilised. The welding parameters for FSW Al–Cu joints were maintained at their working ranges. For multiple response factor optimisation, the combinatory performance of the response factors defines the overall desirability function. The desirability factor is the conversion of the measured response values into a dimensionless evaluation of performance in relation to the importance or significance of the factors [36]. The point on each plot shown in Figure 8 represents the values that fulfil all the prescribed welding process criteria and the values for optimised welding parameters and responses. The multi-objective optimisation performed indicates that the Al–Cu joints produced at a rotational speed of 986 r/min, welding speed of 8.6 mm/min, and 35% shoulder overlap ratio generated an optimised shear strength of 5850 kPa and peel strength of 750 kPa with an overall desirability function of 0.94.

### 4.2. Effect of FSW Process Parameters on Cladded Al–Cu Microstructure

Figure 7a,b indicate the perturbation plots accelerated when the tool rotational speed increased from 825 r/min to 1055 r/min. The values for $\sigma_s$ and $\sigma_p$ factors reached to the maximum level, and the plots gradually decrease when high rotational speeds (1320 r/min) were used. At low rotational speed (825 r/min), insufficient heat input was observed to the Al–Cu alloy due to low friction [43]. Insufficient heat input reduced the plastic flow and developed cavities at the Al/Cu interface or in the stirred zone (SZ). Thus, it reduced the $\sigma_s$ and $\sigma_p$ values. By contrast, a high rotational speed of 1320 r/min with turbulence stirring increased the flow of plasticised Al material into the Cu alloy, thereby causing the formation of microdefects also mentioned by Bisadi et al. [44]. Moreover, severe stirring at high

rotational speed led to grain coarsening, resulting in the reduction in $\sigma_s$ and $\sigma_p$ values [45]. Notably, the use of excessively high or low welding speeds significantly affected the Al–Cu interface bonding.

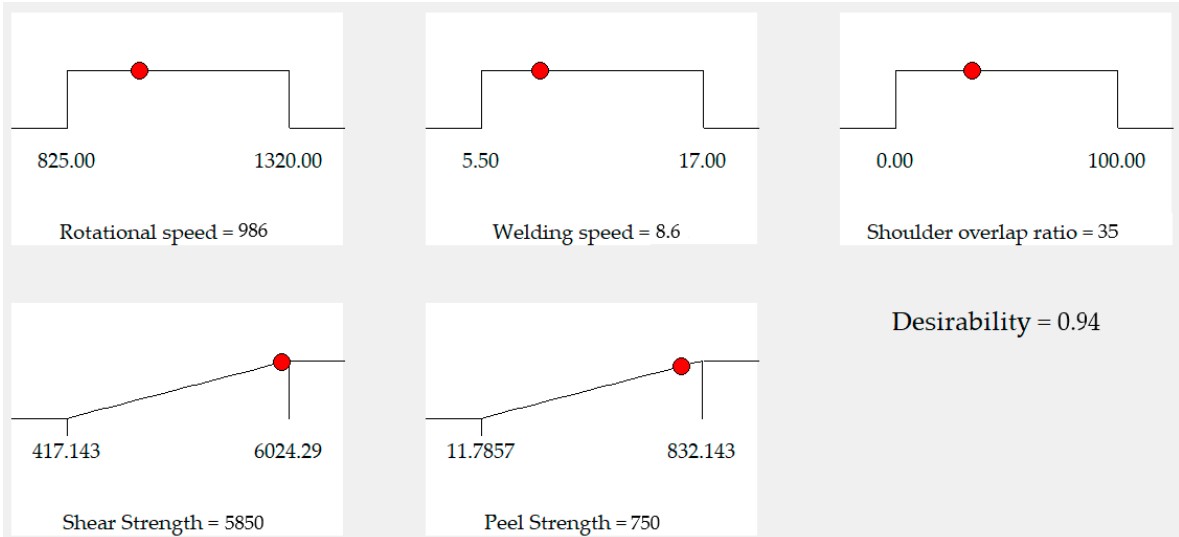

**Figure 8.** The desirability factor and the optimised process parameters.

Figure 7a,b show that the $\sigma_s$ and $\sigma_p$ factors attained the highest value when the welding speed increased from 5.5 mm/min to 9 mm/min before they started to decline at high welding speeds (up to 17 mm/min). At a low welding speed of 5.5 mm/min, the excessive heat was sufficient to promote grain coarsening and the formation of thick intermetallic compounds (IMCs) at the Al/Cu interface. A similar observation was reported by Esmaeili et al. [46] in their study on dissimilar FSW of aluminium 1050 to brass (CuZn30). The formation of IMCs resulted in low Al–Cu interface strength. By contrast, a high welding speed of 17 mm/min led to low heat input and short duration for Al–Cu diffusion and subsequent Al–Cu bonding [47], thereby generating low $\sigma_s$ and $\sigma_p$ values.

During stirring, the shoulder-influenced zone formed when the pin was positioned at a distance for consecutive passes, in which the Al–Cu interface underwent diffusion bonding in this region. Meanwhile, a shoulder overlap ratio of 100% led to a pin overlap area of 100%, thereby producing a pin-influenced zone. In this zone, the Al sheet became plasticised and was passed into the Cu alloy sheet. This effect caused the formation of surface grooves as a result of the taper pin shape, as shown in Figure 6. In addition, a saw-tooth shape (Figure 9) was observed on the surface due to the alternate passes of the tool pin at the shoulder-influenced regions. During this process, a wavy interface was found on samples with material flow toward the advancing side and spiral-shaped zones at the low Cu side. Lakhsminarayanan and Annamalai [48] previously reported that the strength of friction stir clad joints mainly depends on the volume of mixture alloy, which flows into the pin-influenced area, the thickness of IMCs and mechanical interlocking of the cladded interface. The macrostructural and overlap appearances are shown in Figure 9.

Significantly weak Al/Cu interface bonding was observed in the shoulder-influenced zone when a 0% shoulder overlap was employed during cladding. Notably, a maximum shoulder overlap at 100% could intensify liquation, IMC reposition and grain coarsening (Figure 10) as a result of the preheating effects from the preceding pass, which increased the temperature at the intersecting zone. Therefore, the combination of these factors decreased the bonding strength of the Al–Cu joint interface. Nevertheless, the use of an overlap ratio of 50% in cladded joints yielded excellent shear and peel strength compared with the other overlap ratio.

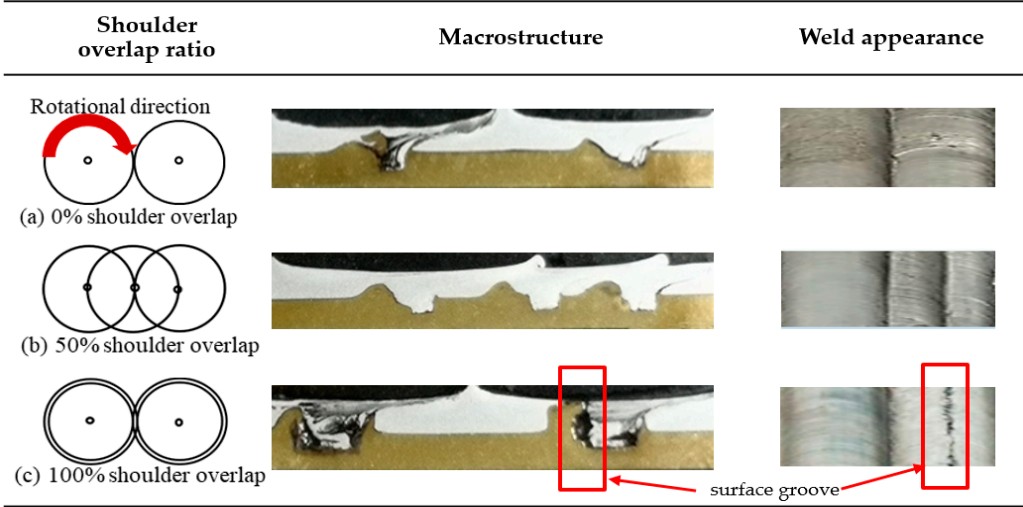

**Figure 9.** Influence of shoulder overlap ratio at Al–Cu clad joined.

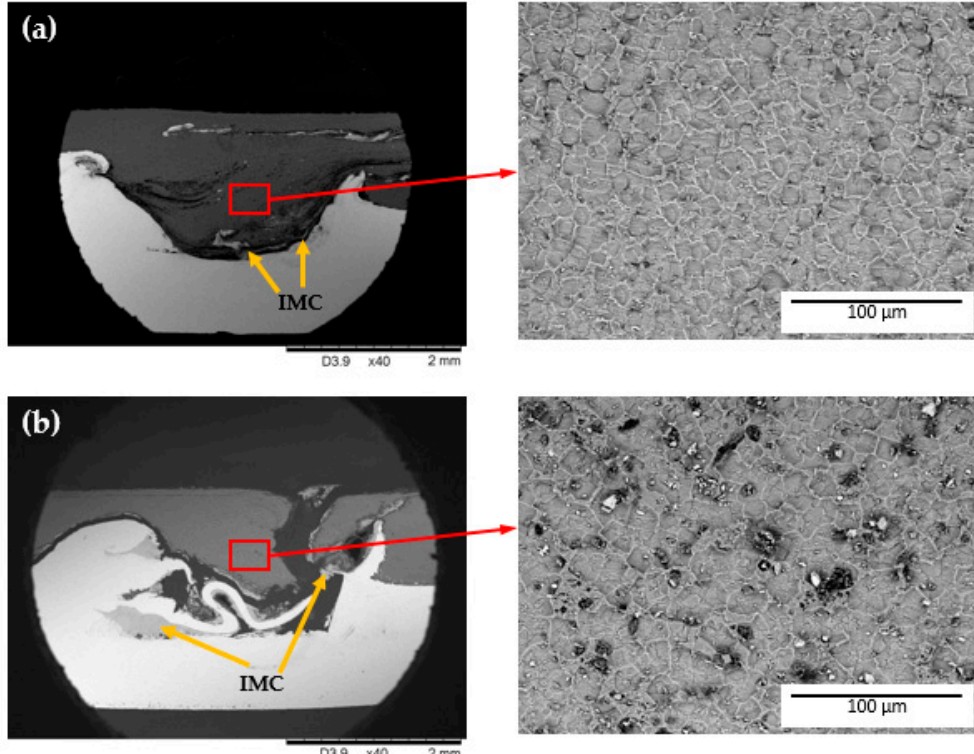

**Figure 10.** Scanning electron microscope (SEM) image of stirred zone (SZ) for (**a**) sample 7 [0% overlap, first pass] and (**b**) sample 25 [100% overlap, first and second pass].

Figure 11 shows the microstructure of as-received Cu and Al with the average grain size of 22 µm and 19 µm, respectively. The microstructure of the as-received AA6061-O comprises $\alpha$-grains containing non-uniformly dispersed $Mg_2Si$ particles. Further analysis was performed on the defect-free sample of cladded Al–Cu joint (sample No. 14 with 50% overlap) to investigate the microstructural changes in different welding zones, as shown in Figures 12 and 13. Figures 12a and 13a shows the defect-free macrostructure of the Al–Cu cladded joint underwent three passes of FSW.

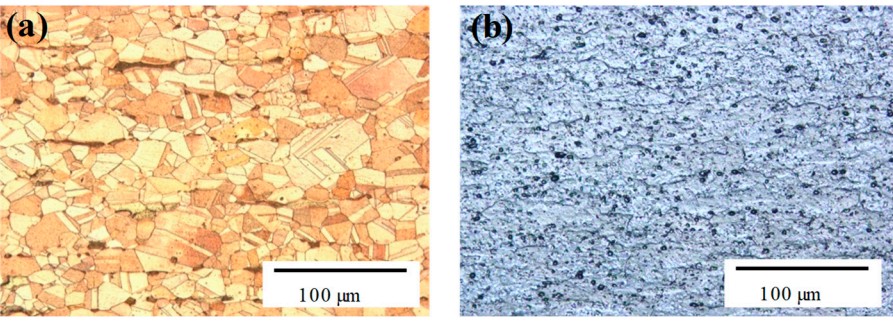

**Figure 11.** The microstructure of as-received metals; (**a**) C2801P alloy and (**b**) AA6061-O alloy.

The FSW of Al–Cu microstructure comprised an SZ, thermo-mechanical affected zone (TMAZ) and heat-affected zone (HAZ) in the welding region. In the Cu side, the zone adjacent to the SZ was TMAZ/HAZ, which experienced deformation that caused grains to be elongated, as illustrated in Figure 12b–g. The SZ in the Al side had an average grain size of 6 μm with small and homogeneous $Mg_2Si$ particles, as shown in Figure 13b,d,f. TMAZ/HAZ in the Al side had an average grain size of 14 μm with large and uniformly distributed $Mg_2Si$ particles, as shown in Figure 13c,e,g.

On the basis of the FESEM and EDS line analyses shown in Figure 14, the pin-influenced region at a 50% overlap ratio produced a stable and strong metallurgical bonding region with Al wt% ranging from 8% to 21%. This zone contributed to the enhancement of recent study by Khodir et al. [47] who reported that the strong metallurgical bonding of Al–Cu with Al alloy content ranging from 10 wt% to 23 wt% further improves the mechanical properties of the joint. The development of these structures shows that the high diffusion rate of Al atoms into Cu side will result in strong metallurgical bonding at the Al–Cu interfaces.

EDS analyses for the cladded structures showed that IMCs have been formed at the Al–Cu interface. X-ray diffraction revealed that the IMCs comprised $Al_2Cu$, and $Al_4Cu_9$ (sample 14 at 50% overlap) as shown in Figure 15. Saeid et al. [36] reported that welding integrity is determined by the type of IMCs formed and 'cold weld' conditions. Therefore, the welding parameters should be adjusted accordingly for optimal joint strength.

The welded fracture surfaces for the sample with thick IMCs were examined to understand their fracture mechanism. Figure 16a,b presents the failure position and fracture surface of sample 26. The failure starts when a crack developed from the hook and propagates inward along with the thick IMC layers at the Al–Cu interface between the SZ and the Cu before moving upward to cut through the SZ. The SEM image illustrates the flat fracture surface which is typical of brittle failure due to intermetallic compound layer as shown in Figure 16b.

## 5. Conclusions

A dissimilar joint of AA6061 aluminium alloy and C2801P copper alloy was successfully achieved by using a multi-pass FSW technique. Al–Cu clad joints exhibited inferior bonding at extremely high rotational and low welding speeds due to defect formation caused by the development of thick IMCs at the Al–Cu interface and grain coarsening. By contrast, the use of extremely low rotational and high welding speeds led to poor Al–Cu bonding due to decreased heat input.

Furthermore, shoulder overlap ratio had a vital influence on the cladded Al–Cu metallurgical bonding, whereas a large gap at 0% overlap yielded a reduction in Al–Cu interface integrity due to the inadequate metallurgical joint between Al and Cu in the shoulder-influenced region. By contrast, an excessive overlap ratio increased the peak temperature of overlying areas as a result of the preheating effect caused by the pin-influenced region of the preceding pass. The increase in temperature caused grain coarsening and thickening of the IMCs formed. In addition, materials processed at a 50% overlap showed outstanding strength due to the development of the mechanical interlocking feature of the cladded interface and the formation of the substantial metallurgical bonding area.

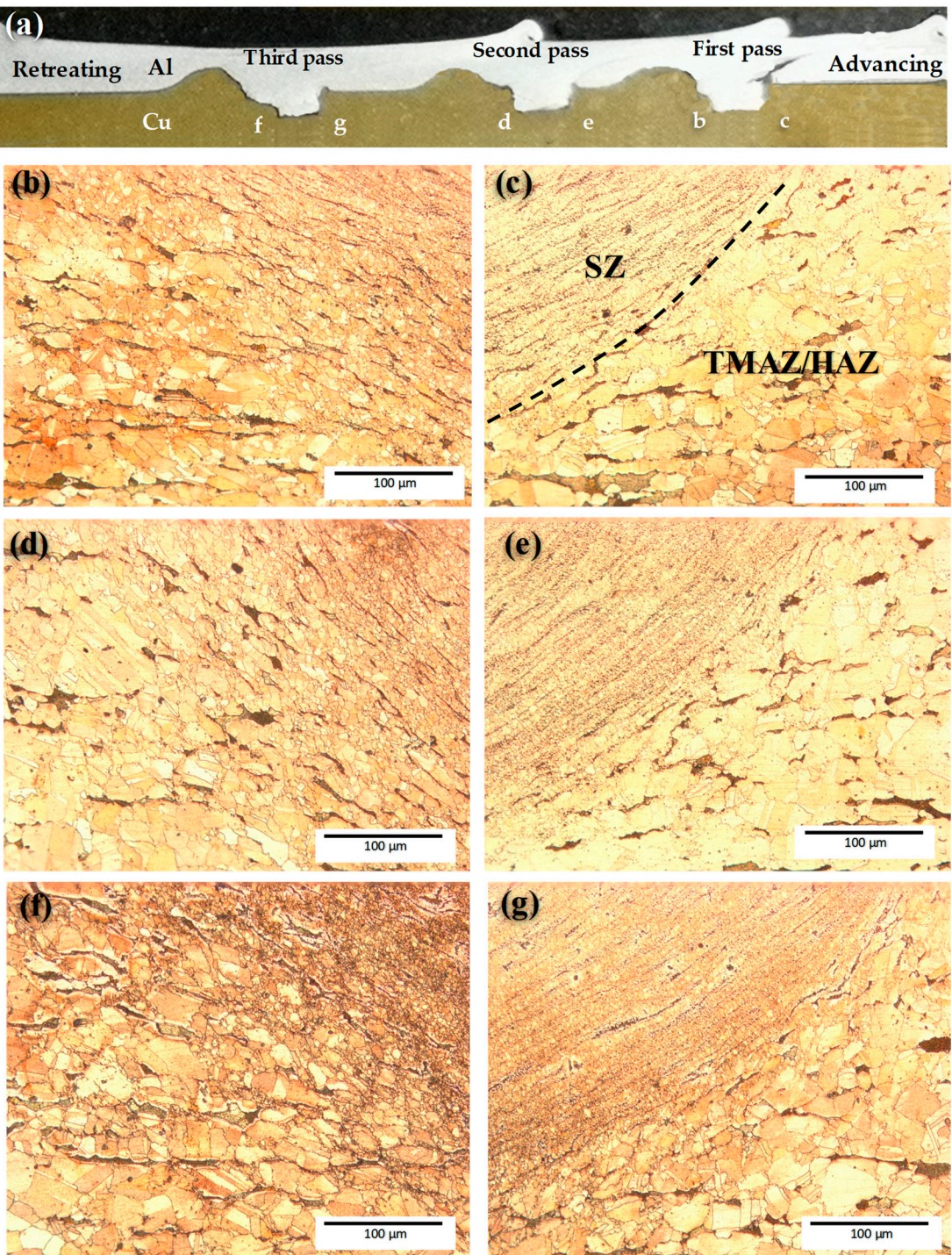

**Figure 12.** Microstructural morphology of Cu alloy after joining; (**a**) macrostructure of friction stir-cladded sample at the condition of 1055 r/min, 9 mm/min, and 50% overlap; (**b**), (**d**) and (**f**) thermo-mechanical affected zone/heat-affected zone (TMAZ/HAZ) of retreating side for the first, second and third passes, respectively, (**c**), (**e**) and (**g**) TMAZ/HAZ of advancing side for the first, second and third passes, respectively.

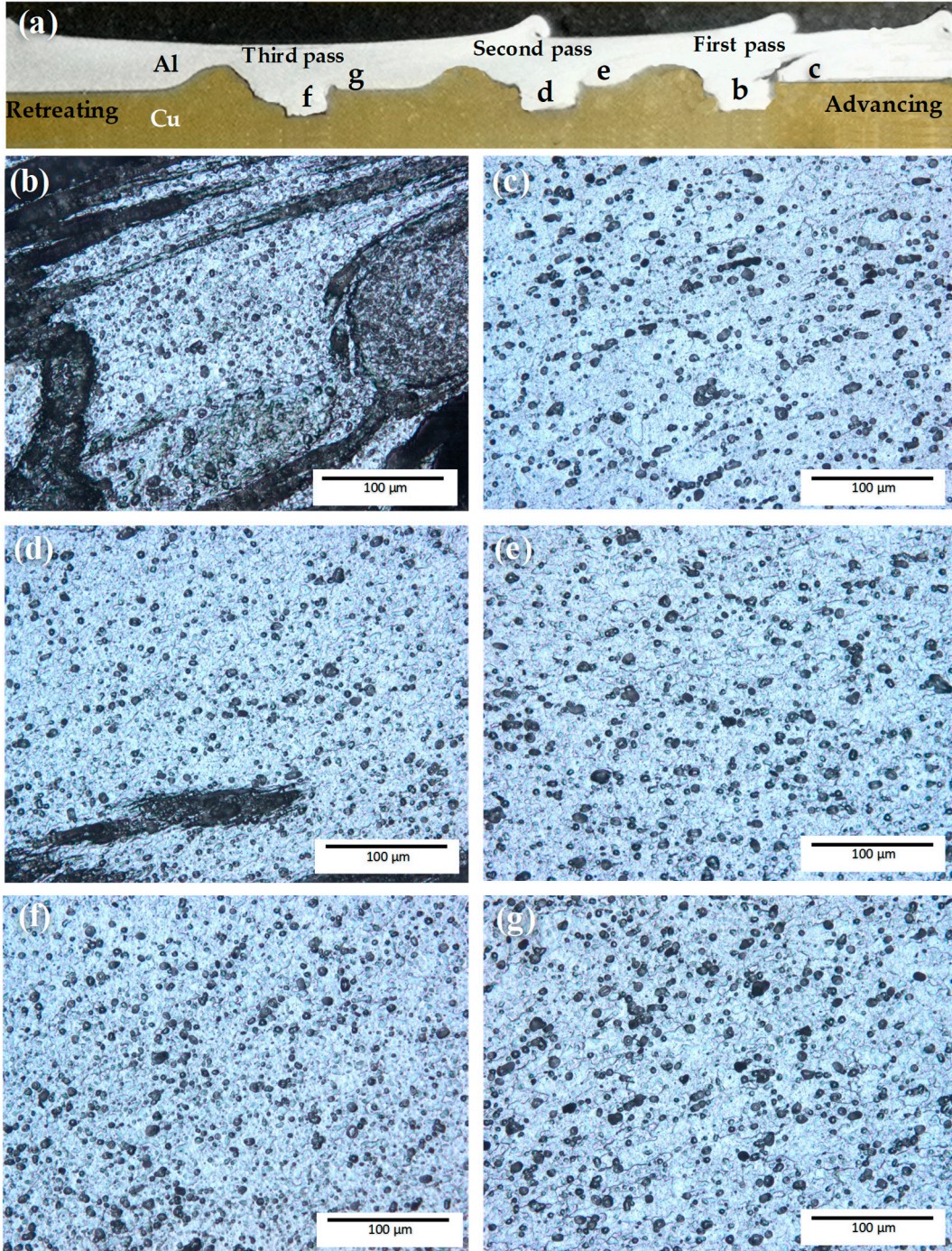

**Figure 13.** Microstructural morphology of Al alloy after joining; (**a**) macrostructure of friction stir cladded sample at the condition of 1055 r/min, 9 mm/min, and 50% overlap; (**b**), (**d**) and (**f**) SZ of the first, second and third passes, respectively, (**c**), (**e**) and (**g**) TMAZ/HAZ of advancing side for the first, second and third passes, respectively.

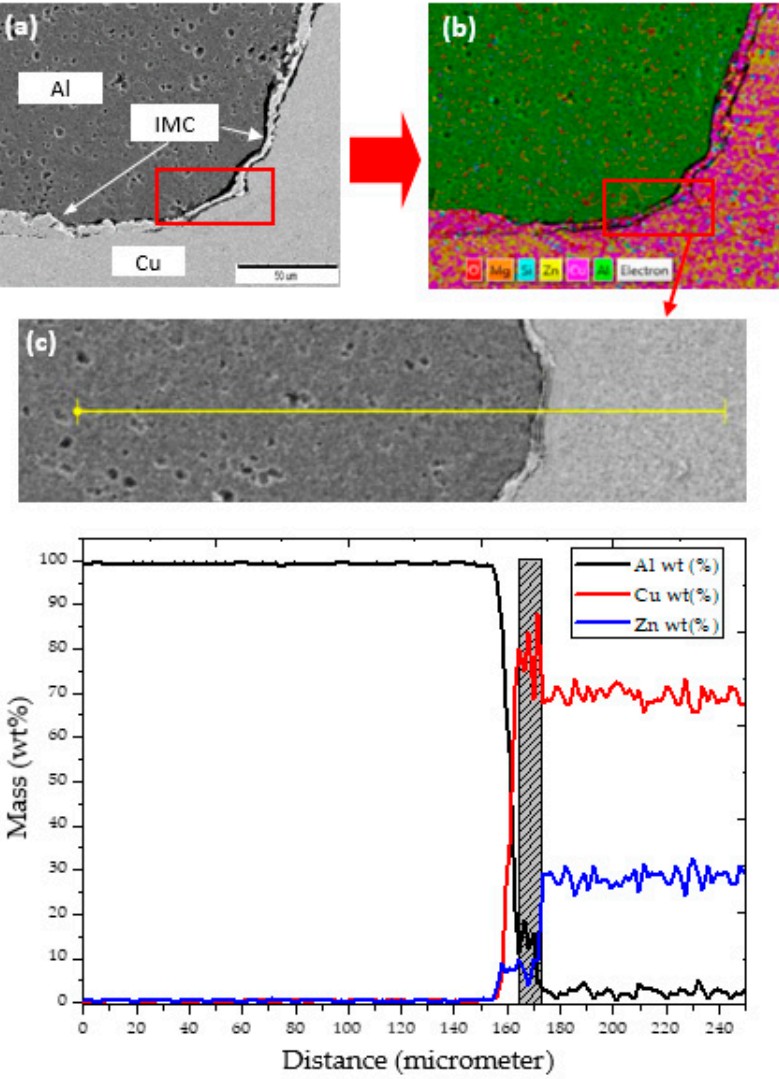

**Figure 14.** Material characterisation analysis for sample 14, (**a**) field-emission scanning electron microscopy (FESEM) analysis, (**b**) energy-dispersive X-ray spectroscopy (EDS) mapping analysis, and (**c**) EDS line scanning analysis.

As a final point, the optimisation outcomes indicated that the cladded Al/Cu is best produced at a rotational speed of 986 r/min, welding speed of 8.6 mm/min and shoulder overlap ratio of 35%. The cladded Al–Cu generated a shear strength of 5850 kPa and peel strength of 750 kPa with an overall desirability function of 0.94.

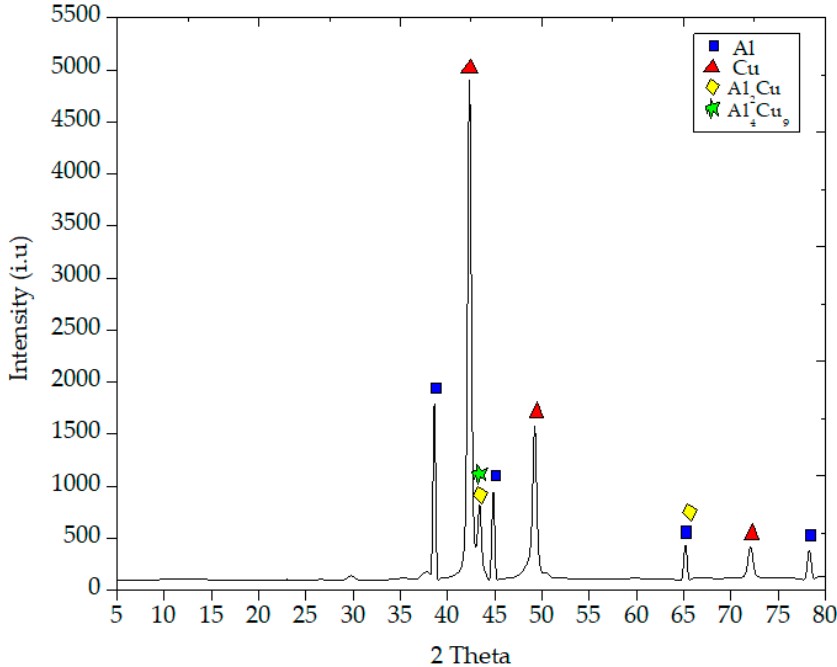

**Figure 15.** X-ray diffraction (XRD) results from the cross-section of Al–Cu clad [Sample 14].

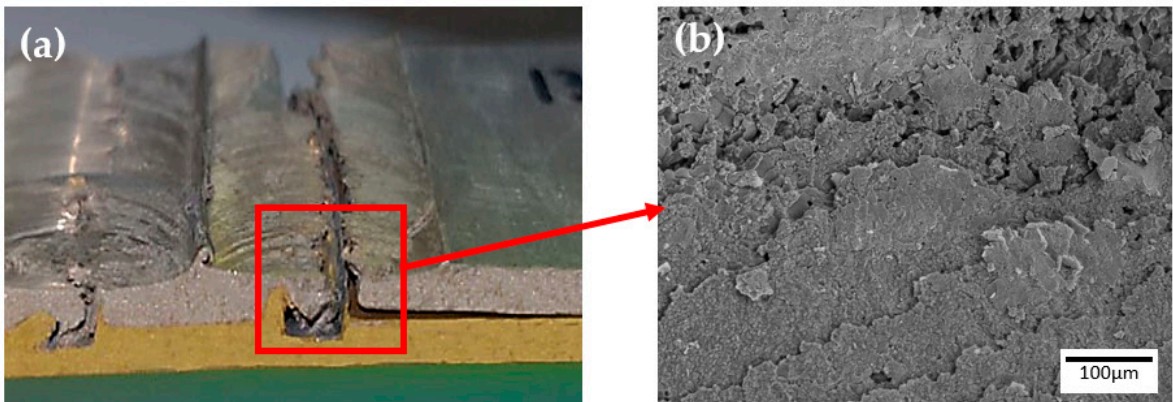

**Figure 16.** Fracture of the cladded Al–Cu of sample No. 26; (**a**) failure position and (**b**) SEM image of the fracture surface.

**Author Contributions:** Conceptualization, Z.S. and N.O.; methodology, N.O. and Z.S.; formal analysis, Z.S. and N.O., investigation, N.O. and Z.S.; writing—original draft preparation, N.O.; writing—review and editing, N.O., Z.S., and A.H.B.; visualization, N.O., Z.S. and A.H.B.; supervision, Z.S. and M.Z.O.; project administration, Z.S. and M.Z.O.

**Funding:** This research was funded by Universiti Kebangsaan Malaysia under the research grant DIP-2014-024.

**Conflicts of Interest:** The authors declare no conflict of interest.

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
