# Peer review of "Effect of Process Parameters on Interfacial Bonding Properties of Aluminium–Copper Clad Sheet Processed by Multi-Pass Friction Stir-Welding Technique"

_metals, doi:10.3390/met9111159_

Round 1

Reviewer 1 Report

The paper was divided into two main sections, experimental and theoretical. The aim was to predict the most optimal parameter combination that can maximize the mechanical response of the joint. Overall, the approach adopted is clearly described but some details need further explanations. Generally, the paper reads well. It can be published in the journal after a few corrections.

Some detailed comments are provided below.

Looking at the results obtained by using two border sets of parameters (voids, rough welds), a search of optimal ones in the narrower range is justified. The authors do not explain the ANOVA method which was used to predict the parameters. Also the P and F variables were not described. The optimized parameters presented in Figure 11 should allow obtaining the clad joint with very high strength. However, do the authors have any experimental data to verify those predictions from the simulation?

Reviewer 2 Report

I read the manuscript 611914 and pointed out some questions as follows;

1. Line 100; “The tensile-shear test” must be change to “The shear test”. Because “the tensile test” and “the shear test” are different test each other from the engineering design view point. Similarly, the expression of “tensile shear” should change to “shear” throughout this manuscript. The title of ASTM D1002 standard cited by authors is “Standard Test Method for Apparent Shear Strength of Single-Lap-Joint Adhesively Bonded Metal Specimens by Tension Loading (Metal-to-Metal)”. This means “shear strength by tension loading”, so don’t mix shear test and tensile test.

2. Line 100-104: What is the size of both peel and shear tests? Please show by additional Figures not only mention ASTM standards. Because reader cannot estimate the shear stress from the shear force Fs that you indicate Table 4. I mean the shear stress can be calculated by Fs/(Area of shear test) and used for safety design.

3. Figure 1; Please specify the plate thickness in the figure. Total thickness = 3mm, thickness of Al and Cu are 1.5mm respectively.

4. Figure 2; Enter specific values of overlap by (%), for example 0%, 50% and 100%.

5. Line132-134 and Table 3; I think that 3 level is not enough to decide the optimal values of w, v and Sr, because the effects of parameters is not always proportional to Fs and Fp. Please show authors opinion.

6. Figure 4(a) and Figure 4(b); The vertical and horizontal axes of these figures are mentioned only “Predicted” and “Actual”. Authors had better add Fs(N) and Fp(N) as unit of values.

7. Line245-246; Please point out the evidences of “grain coarsening and the formation of IMCs”.

8. Line307-308; When authors want to wire the existence of CuAl2 and Cu9Al4, they should mention the X-ray diffraction patterns of each IMCs in this manuscript.

9. Figure 10(c); Please color this figure, because readers cannot understand which line is Al, Cu or Zn.

10. Conclusions Line341; If authors conclude strength of Al-Cu clad is effected by thickness of IMCs at the Al/Cu interface, grain coarsening and poor Al/Cu bonding. I think readers need a directly evidence of “Fractography” that show the interaction between fracture and IMCs. In other word, authors had better indicate the photograph of typical fracture surface and cracking propagation pass and existence of IMCs by using the SEM and EDS in this manuscript before conclusion.

that's all.

Reviewer 3 Report

The manuscript presents the results of a process parameter study for FSW of Al to Cu substrates. However, this reviewer finds the methods analysis and presentation of the results to be critically flawed and cannot recommend publication.

The following critical points are highlighted, but are not exhaustive:

From the outset it is unclear what is the cladding material and what is the substrate, only by inference from the figures can it be deduced that Al is the cladding. There are (absolutely) no details about the specimens that were made to perform the tensile-shear and peel tests that are essential for the success of this study. All details need to be provided, including specimen size, where and how cut from the welded 150x150mm blanks, number of specimens tested for each parametric variation, stats on the results, etc. From table 4 it appears that only 1 test was done for each set of variables. This cannot provide a statistically significant result, a minimum of 3 tests should be performed for each st of variables. It is not meaningful to express the strength results in terms of load, even worse since the size of the specimen is not stated. Values should be given in stress units. In some places the text talks about shear stress, but only load values are given. The peel loads (210N, for example) are not likely valid or significant if measured on a 100kN load frame. That value is 0.2% of the load cell capacity. A load cell with much lower capacity is necessary. Shear loads of 4000N are at the low end of any chance of being significant. It is unreasonable to state the optimum parameters including rotational speed and transverse speed with 3-5 significant figures: 988.86 r/min, welding speed of 8.87 mm/min. This indicates a lack of understanding of the limitations, and interpretation of a process parameter study.

For these reasons alone, the recommendation is not to publish this work in this journal.

Round 2

Reviewer 2 Report

Good. The revised color version of manuscript 611914 include all information what I want to know. Authors carefully revised following to my all questions. (like a Japanese). I think as a result; the revised manuscript has increased both scientifically and engineering very much. I acknowledge that this manuscript 611914 is worth publishing as a scientific and engineering paper.

Reviewer 3 Report

The authors still do not appreciate the invalidity of their analysis and conclusions.

I appreciate the effort to improve the manuscript but cannot recommend publication in its present from. Until the following are corrected (it appears in the Abstract, Discussion and Conclusions) the work is not credible.    

"the optimisation outcomes indicated that the cladded Al/Cu is best produced at  a rotational speed of 986.85 r/min, welding speed of 8.57 mm/min and shoulder overlap ratio of 406 34.91%. The cladded Al/Cu generated a shear strength of 5845.8 kPa and peel strength of 754.136 kPa  with an overall desirability function of 0.936."

In three locations the number of significant digits claimed for a optimized process is not reasonable, bordering on ridiculous.

It is suggested that 986 rpm, 8.6 mm/min 5850 kPa and 750 kPa would be appropriate values. To insist on the implied precision claimed in the original and revised manuscripts goes against scientific credibility.

Round 3

Reviewer 3 Report

The manuscript is now recommended for publication.